# Human osteoclastogenesis in Epstein-Barr virus-induced erosive arthritis in humanized NOD/Shi-*scid/IL-2Rγ*null mice

Yosuke Nagasawa[1,ORCID,◎], Masami Takei[1,◎]*, Mitsuhiro Iwata[1], Yasuko Nagatsuka[1], Hiroshi Tsuzuki[1], Kenichi Imai[2], Ken-Ichi Imadome[3], Shigeyoshi Fujiwara[1,4,ORCID], Noboru Kitamura[1]*

**1** Division of Hematology and Rheumatology, Department of Medicine, Nihon University School of Medicine, Tokyo, Japan, **2** Department of Microbiology, Nihon University School of Dentistry, Tokyo, Japan, **3** Department of Advanced Medicine for Infections, National Center for Child Health and Development, Tokyo, Japan, **4** Department of Allergy and Clinical Immunology, National Research Institute for Child Health and Development, Tokyo, Japan

◎ These authors contributed equally to this work.
* takei.masami@nihon-u.ac.jp (MT); kitamura.noboru@nihon-u.ac.jp (NK)

**Data Availability Statement:** All relevant data are within the paper and its Supporting information files.

## Abstract

Many human viruses, including Epstein-Barr virus (EBV), do not infect mice, which is challenging for biomedical research. We have previously reported that EBV infection induces erosive arthritis, which histologically resembles rheumatoid arthritis, in humanized NOD/Shi-*scid/IL-2Rγ*null (hu-NOG) mice; however, the underlying mechanisms are not known. Osteoclast-like multinucleated cells were observed during bone erosion in this mouse model, and therefore, we aimed to determine whether the human or mouse immune system activated bone erosion and analyzed the characteristics and origin of the multinucleated cells in hu-NOG mice. Sections of the mice knee joint tissues were immunostained with anti-human antibodies against certain osteoclast markers, including cathepsin K and matrix metalloproteinase-9 (MMP-9). Multinucleated cells observed during bone erosion stained positively for human cathepsin K and MMP-9. These results indicate that human osteoclasts primarily induce erosive arthritis during EBV infections. Human osteoclast development from hematopoietic stem cells transplanted in hu-NOG mice remains unclear. To confirm their differentiation potential into human osteoclasts, we cultured bone marrow cells of EBV-infected hu-NOG mice and analyzed their characteristics. Multinucleated cells cultured from the bone marrow cells stained positive for human cathepsin K and human MMP-9, indicating that bone marrow cells of hu-NOG mice could differentiate from human osteoclast progenitor cells into human osteoclasts. These results indicate that the human immune response to EBV infection may induce human osteoclast activation and cause erosive arthritis in this mouse model. Moreover, this study is the first, to our knowledge, to demonstrate human osteoclastogenesis in humanized mice. We consider that this model is useful for studying associations of EBV infections with rheumatoid arthritis and human bone metabolism.

**Funding:** This work was supported by the MEXT-Supported Program for the Strategic Research Foundation at Private Universities and Nihon University Multidisciplinary Research Grant for 2017. The funders had no role in study design, data collection and analysis, decision to publish, or preparation of the manuscript.

**Competing interests:** The authors have declared that no competing interests exist.

## Introduction

Environmental factors, including infectious agents, contribute to the pathogenesis of autoimmune diseases along with genetic factors. The Epstein-Barr virus (EBV), a human herpesvirus infecting > 90% of the global adult population, is associated with infectious mononucleosis, lymphoproliferative disorders of immunocompromised hosts, Burkitt's lymphoma, and nasopharyngeal carcinoma. In addition, EBV is suggested to be associated with autoimmune diseases, such as rheumatoid arthritis (RA) [1–9], Sjögren's syndrome [2,10–12], systemic lupus erythematosus [13–15], autoimmune thyroid disorders [16,17], and multiple sclerosis [18,19]. Evidence indicating the possible involvement of EBV in the pathogenesis of RA includes higher circulating EBV load and B-cell responses to the virus in patients with RA than in controls, aberrant T-cell responses to EBV in patients with RA, and the existence of EBV proteins and nucleic acids in RA synovial tissues. Molecular mimicry between several EBV proteins and cellular antigens of synovial components has also been documented [20]. Furthermore, EBV-infected plasma cells producing antibodies to citrullinated peptides were recently detected in the synovial lymphoid structures of RA [21].

As EBV infects only humans and only a few species of the New World monkeys under experimental conditions, development of an appropriate animal model to prove a causal relationship between EBV and human diseases associated with the virus, including RA, has been difficult. Recently, the components of the human immune system in immunodeficient mice, such as NOD/Shi-scid/IL-2Rγnull (NOG), were reconstituted by transplanting human CD34$^+$ hematopoietic stem cells (HSCs) in mice. These mice are referred to here as humanized NOG (hu-NOG) mice [22]. In hu-NOG mice, most major components of the human immune system, including T cells, B cells, natural killer (NK) cells, monocytes/macrophages, and dendritic cells, were reconstituted, and upon infection with EBV, these mice reproduced the key aspects of human EBV infection, including innate and adaptive immune responses [23,24]. We have previously reported that EBV-infected hu-NOG mice develop erosive arthritis with histological features of RA, such as massive synovial proliferation, bone erosion, and bone marrow edema [25]. In addition, a pannus-like structure formed by massive synovial proliferation is a particularly characteristic feature of these mice. This pannus-like granulated tissue invaded the bone surface and caused bone erosion. Furthermore, osteoclast-like multinucleated giant cells lined the erosion zone. However, the incidence rate of erosive arthritis was not high, and only histological assessment was used for evaluation. Furthermore, the molecular mechanisms underlying erosive arthritis in these mice have not been elucidated.

This study aimed to determine whether the human or mouse immune system induced bone erosion in EBV-infected hu-NOG mice, with particular emphasis on the origin of osteoclasts inducing erosive arthritis. Thereafter, we analyzed the conditions under which erosive arthritis occurred at a high rate, focusing on the elevation in the levels of CD8$^+$ peripheral blood lymphocytes (PBLs) after EBV infection. We speculate that EBV infections contribute to some of the unclear factors influencing RA pathogenesis, and we believe that these EBV-infected hu-NOG mice constitute an erosive arthritis model, potentially yielding insights into RA pathogenesis.

## Materials and methods

### Generation of hu-NOG mice

NOG mice were obtained from the Central Institute for Experimental Animals (Kanagawa, Japan). The protocols for experiments with NOG mice were approved by the Institutional Animal Care and Use Committees of Nihon University (certification number, AP13M041) and by

the Nihon University Biosafety Committee for Gene Recombination (certification number, 2006-M). 31 mice were prepared for the experiment, health checkups were performed thrice a week, and weight measurements were taken once a week. In order to reduce the pain of the mice, the experiment was planned such that the timing of euthanasia coincided with the time when the mice exhibited anguish symptoms or significant weight loss (20% or more per week) during the course of the experiment. Seven-week-old female NOG mice were transplanted with human cord blood $CD34^+$ HSCs (2C-101, Lonza, Basel, Switzerland) at a rate of 8.0– $10 \times 10^4$ cells/mouse via the tail vein. The reconstituted human immune system components were evaluated and characterized by monitoring the percentages of human $CD3^+$, $CD4^+$, $CD8^+$, $CD19^+$, and $CD45^+$ cells in mouse PBLs using flow cytometry.

## Preparation of EBV and infection of hu-NOG mice with EBV

The protocols for our experiments with EBV were approved by the Biorisk Management and Control Committee of Nihon University School of Medicine (certification number, 20-13-5). For preparing EBV, cells of the EBV-producer line B95-8 (JCRB9123, Japanese Collection of Research Bioresources, Osaka, Japan) were cultured with Roswell Park Memorial Institute (RPMI)-1640 medium (Sigma Aldrich, St. Louis, MO, USA), supplemented with penicillin G (100 U/mL), streptomycin (100 μg/mL), and 10% fetal bovine serum (FBS) at 37˚C in a 5% $CO_2$ incubator. The culture supernatant of the B95-8 cells was collected 7 days after the final medium change. The supernatant was removed after centrifugation at $400 \times g$ for 5 min at 4˚C. The enriched virus fluid was filtered through a 0.45-μm membrane and stored at −80˚C.

For titrating EBV, donor cord blood samples were obtained with written informed consent of the donor's parents. The protocols for our titration experiments with donor cord blood samples were approved by the Nihon University Itabashi Hospital Clinical Research Judging Committee (certification number, RK-140613-11). Mononuclear cells isolated from cord blood were plated at a density of $2.0 \times 10^5$ cells/well in 96-well plates and then inoculated with serial 10-fold dilutions of the virus preparation. The number of wells with clumps of proliferating cells was counted 6 weeks after infection, and the titer of the virus in 50% transforming dose ($TD_{50}$) was determined using the Reed-Muench method [26]. After 3–4 months of human HSC transplantation, the hu-NOG mice were inoculated with EBV at a dose of 1.0– $2.0 \times 10^1$ $TD_{50}$/100 μL/mouse via the tail vein.

## Flow cytometry

PBLs isolated from EBV-infected and -uninfected hu-NOG mice were stained with the following monoclonal antibodies (IgG1 subtype): ECD (R phycoerythrin (PE)-Texas Red®-X)-conjugated anti-human CD3 (A07746, UCHT1, Beckman Coulter, Marseille, France), PE-conjugated anti-human CD4 (561842, RPA-T4, Becton Dickinson, Franklin Lakes, NJ, USA), fluorescein isothiocyanate (FITC)-conjugated anti-human CD8 (555636, HIT8a, Becton Dickinson), PC7 (R PE-cyanine 7)-conjugated anti-human CD19 (IM2708U, J3-119, Beckman Coulter), and peridinin chlorophyll A protein/cyanine 5.5 (PerCP/Cy5.5)-conjugated anti-human CD45 (304001, HI30, BioLegend, San Diego, CA, USA). Mouse IgG1 conjugated to a fluorescent dye corresponding to each monoclonal antibody was used as the negative control. All stained cells were analyzed using multicolor flow cytometry with the FC500 flow cytometer (Beckman Coulter). Typically, live lymphocytes, determined through forward and side-scatter parameters, were gated for analysis.

## Histochemistry of knee joint tissues

EBV-infected and -uninfected hu-NOG mice were sacrificed via cervical subluxation by trained technicians. After confirmation of death, the knee joints were dissected out from these animals, fixed in 10% formaldehyde solution, and embedded in paraffin. Serial sections were generated along the longitudinal bone axis from the paraffin-embedded samples and subjected to staining for tartrate-resistant acid phosphatase (TRAP) and with hematoxylin-eosin. The sections of EBV-infected and -uninfected hu-NOG mice were stained for human cathepsin K and human matrix metalloproteinase-9 (MMP-9) using immunohistochemistry.

## Examination of mouse knee joint using three-dimensional computed tomography

EBV-infected and -uninfected hu-NOG mice were euthanized and three-dimensional images of the knee joints were constructed from multiple tomographic images using a high-definition microfocus X-ray computed tomography scanner (Kureha Special Laboratory Co., Ltd., Fukushima, Japan).

## Bone marrow cell culture

Osteoclasts are generally derived from their progenitor cells of the monocyte/macrophage lineage in the bone marrow [27]. Long bones, such as the femur of the EBV-infected and -uninfected mice were dissected and cut off at the epiphysis. The marrow was flushed out with RPMI-1640 medium (Sigma Aldrich) containing 10% heat-inactivated FBS using a 26-gauge needle attached to a 1.0-mL syringe and collected in a 1.5-mL tube. After centrifugation ($200 \times g$ for 15 min at 26˚C), bone marrow cells were obtained from EBV-infected mice, suspended in osteoclast growth medium (PT9501, Lonza) containing recombinant human macrophage colony-stimulating factor (M-CSF) (33 ng/mL) and soluble human receptor activator of nuclear factor κB ligand (RANKL) (66 ng/mL), and supplemented with L-glutamine (0.1%), penicillin G (100 U/mL), streptomycin (100 μg/mL), and 10% FBS. Bone marrow cells from EBV-uninfected mice and commercially available mouse osteoclast progenitor cells (OSC14C, Cosmo bio, Tokyo, Japan) were suspended in (human) osteoclast culture medium (OSCMHB, Cosmo bio) and (mouse) osteoclast culture medium (OSCMM, Cosmo bio), respectively. These bone marrow cells and the mouse osteoclast progenitor cells were then plated into chamber slides (Laboratory-Tek 8-well Permanox Slides, Thermo Scientific/Thermo Fisher, Grand Island, NY, USA) at a density of $1.0 \times 10^4$ cells/well and incubated at 37˚C in a 5% $CO_2$ incubator. After 10–14 days of culture, the chamber slides were subjected to cytochemistry.

Pit formation assay was performed as follows. Bone marrow cell preparations from EBV-infected hu-NOG mice obtained were placed in osteo assay surface plates coated with a synthetic inorganic bone mimetic calcium phosphate (Corning, Kennebunk, ME, USA), which allows direct assessment of osteoclast activity in vitro [28], and incubated with osteoclast growth medium (Lonza) in the presence of recombinant human M-CSF (33 ng/mL) and soluble human RANKL (66 ng/mL) at 37˚C in a 5% $CO_2$ incubator. After 10 days of incubation, the plates were stained for TRAP and examined using light microscopy.

Osteoclasts were separated from bone marrow cells using a previously reported method of culturing osteoclasts in vitro as adherent cells [29]. The pelvic bones of the EBV-infected and -uninfected mice were dissected and cut into several pieces. The marrow was flushed out with RPMI-1640 medium (Sigma Aldrich) containing 10% heat-inactivated FBS and collected in a 1.5-mL tube. The bone marrow cells obtained from pelvic bone were filtered using a 40 μm cell strainer and stored at −80˚C using Cell Banker (Nippon Zenyaku Kogyo Co., Ltd,

Fukushima, Japan) until use. After thawing, the bone marrow cells from pelvic bones were cultured in α-minimal essential medium (MEM) (Gibco/Thermo Fisher, Grand Island, NY, USA) containing penicillin G (100 U/mL), streptomycin (50 μg/mL), gentamicin (50 μg/mL), and 20% FBS overnight at 37˚C in a 5% $CO_2$ incubator to remove adherent cells. After 12 hours, the non-adherent cells were collected and seeded on glass-bottom dishes. After 3 days of culture, the non-adherent cells were washed out with fresh media. After 8 days of culture, the adherent cells were subjected to cytochemistry.

## TRAP staining

The plates of serial sections from the knee joint samples, the chamber slides of cultured bone marrow cells of long bones, the pit formation assay plates of cultured bone marrow cells of long bones, and glass-bottom dishes of cultured bone marrow cells of pelvic bones were subjected to staining for TRAP using the acid phosphatase leukocyte kit (3864-1KT, Sigma Aldrich). The samples were fixed in a citrate/acetone solution. After rinsing in deionized water, they were incubated with a mixture of acetate solution, naphthol AS-BI phosphoric acid solution, tartrate solution, and Fast Red Violet LB salt solution (F3881, Sigma Aldrich) in a dark room. After washing, serial sections in the plates and cultured bone marrow cells of pelvic bones in glass-bottom dishes were stained with hematoxylin (cultured cells on the chamber slides and pit formation assay plates were not stained).

## Immunohistochemistry

The plates containing serial sections of the knee joint samples were immunostained for human cathepsin K and human MMP-9, which are osteoclast markers and are involved in bone resorption [30–33]. After deparaffinization using xylene and drysol, the plates were treated with 0.3% hydrogen peroxide/methanol for blocking. After inactivation of endogenous peroxidase, they were incubated with Background Sniper (BS966, BioCare Medical, Concord, CA, USA) and then incubated with rabbit anti-human cathepsin K polyclonal antibody (M189, Takara, Shiga, Japan; dilution, 1:50) and rabbit anti-human MMP-9 polyclonal antibody (LS-C95901, Life Span Bioscience, Inc, Seattle, WA, USA; dilution, 1:25, 1:50, and 1:100), followed by incubation with horseradish peroxidase-conjugated goat anti-rabbit antibody (Histofine Simplestain, MAXPO(R), 424141, Nichirei Biosciences Inc., Tokyo, Japan). Next, they were incubated with 3,3'-diaminobenzidine (425011, Nichirei Biosciences Inc.) and counterstained with hematoxylin. All stained plates and chamber slides were examined using light microscopy.

## Immunocytochemistry

Chamber slides of cultured bone marrow cells of long bones were immunostained for human cathepsin K, human MMP-9, and human mitochondria, and those of cultured mouse osteoclast progenitor cells were stained for mouse MMP-9, human MMP-9, mouse mitochondria, and human mitochondria. The chamber slides were fixed in methanol and treated with 0.2% Triton X-100 (04605, Polysciences, Warrington, PA, USA). After the endogenous peroxidase was inactivated with peroxidase-blocking solution (S202386-2, Dako, Glostrup, Denmark), they were incubated with Background Sniper (BioCare Medical) for blocking and then incubated with rabbit anti-human cathepsin K polyclonal antibody (Takara; dilution, 1:200, 1:600, 1:1800, and 1:5400), rabbit anti-human MMP-9 polyclonal antibody (Life Span Bioscience, Inc; dilution, 1:12.5, 1:25, and 1:50), rabbit anti-MMP-9 polyclonal antibody (orb13583, Biobyt, Cambrigeshire, UK; dilution, 1:50, 1:100, and 1:200), mouse monoclonal antibody against the 60 kDa non-glycosylated protein component of the human mitochondria (NB600-556,

NBP2-32982, Novus Biologicals, Littleton, CO, USA; dilution, 1:10 and 1:40), rabbit anti-Prohibitin polyclonal antibody (ab28172, Abcam, Cambrigeshire, UK; dilution, 1:180, 1:360, and 1:720), negative control rabbit immunoglobulin (X0936, Dako; dilution, 1:1500, 1:4500, 1:13500, and 1:40500), or negative control mouse immunoglobulin (X0931, Dako; dilution, 1:5 and 1:20), followed by incubation with horseradish peroxidase-conjugated goat anti-rabbit or anti-mouse secondary antibody (K4003, K4001, Dako). Thereafter, the samples were incubated with 3,3'-diaminobenzidine (Nichirei Biosciences Inc.) and counterstained with hematoxylin. All stained chamber slides were examined using light microscopy.

The cultured bone marrow cells from pelvic bones in glass-bottom dishes were fixed in citrate/acetone solution (Sigma Aldrich), incubated with PBS containing 5% FBS, 2% bovine serum albumin, and 0.4% Triton X-100 (Polysciences) for blocking, and then incubated with rabbit anti-human MMP-9 antibody (Life Span Bioscience, Inc, dilution, 1:1000). This was followed by incubation with goat anti-rabbit secondary antibody (DI-1549, DyKight 549, Vector Laboratories, Burlingame, CA, USA). All stained glass-bottom dishes were examined using fluorescence microscopy.

## Statistical analysis

The number of CD34+ cells transplanted between the EBV-infected and EBV-uninfected groups was analyzed using the Mann-Whitney *U*-test. The incidence rate of bone erosion between the EBV-infected and EBV-uninfected groups was compared using Fisher's exact test. The relationship between the severity of bone erosion and titer of EBV inoculated was analyzed using the Mann-Whitney *U*-test. These analyses were performed using Excel for Mac 2011 version 14.7.7 (Microsoft, Redmond, WA, USA). A P value < 0.01 was considered to indicate a statistically significant difference.

## Results

The 31 NOG mice used in this study were humanized, and 21 of these were infected with EBV. There was no mortality except for the planned euthanasia.

### Bone erosion induced by EBV infection

In this study, we initially confirmed our previous findings of rapid increase in the number of CD8+ cells and the reversal of CD4/CD8 ratio in the peripheral blood of humanized mice following EBV infection [24]. The percentages of CD4+ cells and CD8+ cells in the PBLs of EBV-infected (n = 10) and EBV-uninfected mice (n = 5) were sequentially assessed using flow cytometry once per week. In all 5 EBV-uninfected mice, the CD4/CD8 ratio remained above 1.0 during our observation period. In contrast, the CD4/CD8 ratio in EBV-infected mice decreased to less than 1.0 in all 10 EBV-infected mice, reaching minimal values 7–10 weeks after EBV inoculation in most mice. The changes in the percentages of CD4+, CD8+, CD19+ and CD45+ cells in PBLs of EBV-infected mice are shown in Fig 1. As the rapid decrease in the CD4/CD8 ratio indicated successful infection with EBV [24], the histology of the knee joint tissues of mice was analyzed. Breakdown of the bone surface was observed in the area close to the knee joint capsule in all 10 EBV-infected mice, whereas no sign of bone destruction was detected in any of the five EBV-uninfected mice (Table 1). Although the number of CD34+ cells transplanted between the EBV-infected and EBV-uninfected groups did not show any correlation (P = 0.052), the incidence rate of bone erosion was significantly high in the EBV-infected group (P = 0.0003). The histological severity of bone erosion around the knee joint was classified into three grades according to the depth of cortical bone breakdown (CBB): CBB $\geq$ 2/3$^{rd}$ of the bone thickness, 1/3$^{rd}$ $\leq$ CBB < 2/3$^{rd}$ of the bone thickness, and

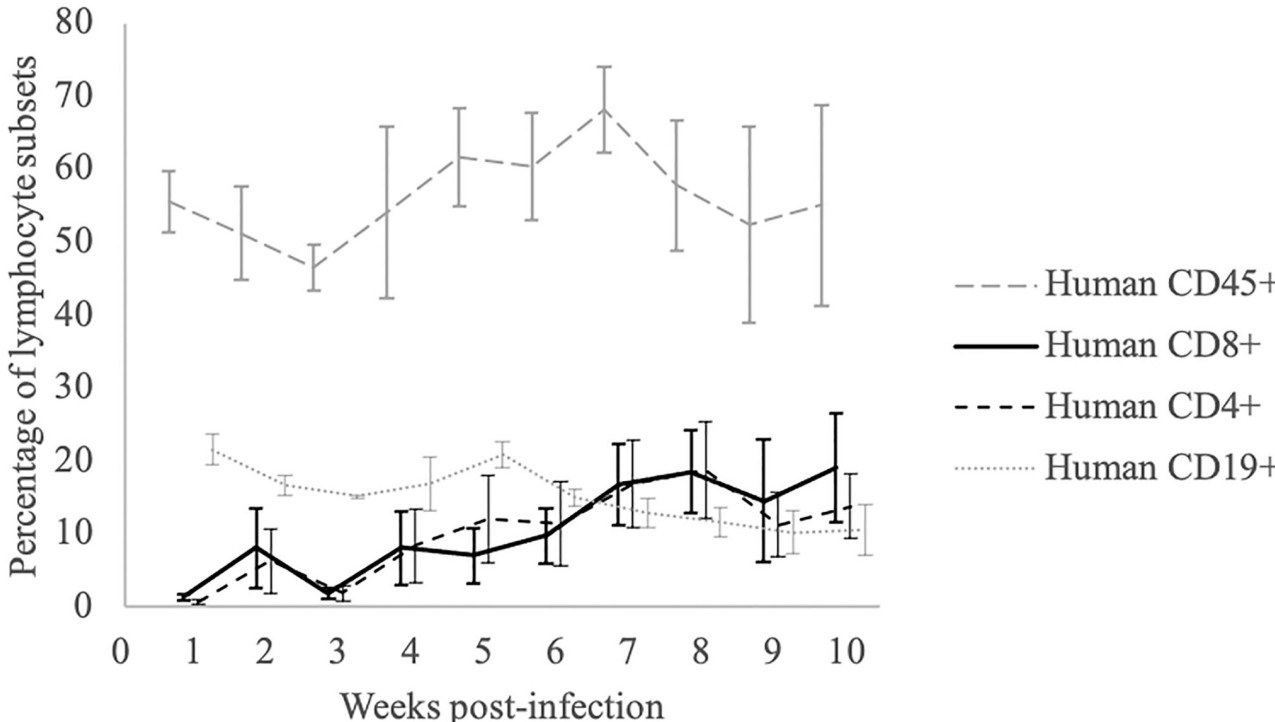

**Fig 1. Time course of human lymphocyte reconstitution in peripheral blood of EBV-infected hu-NOG mice.** Following inoculation with EBV, the percentages of human CD45[+], CD4[+], CD8[+], and CD19[+] cells among the peripheral blood mononuclear cells were measured weekly. Upon EBV infection, the percentage of human CD8[+] T cells increased rapidly, and the ratio of CD4[+] cells to CD8[+] cells decreased to below 1. Values represent mean ± standard error.

**Table 1. Bone erosion in hu-NOG mice.**

|  | Number of transplanted CD34[+] cells | Titer of inoculated EBV (TD$_{50}$) | Severity of bone erosion |
|---|---|---|---|
| EBV-infected mice |  |  |  |
| NOG 1–3 | $1.0 \times 10^5$ | 10 | 3+ |
| NOG 1–9 | $1.0 \times 10^5$ | 10 | 3+ |
| NOG 2–2 | $8.0 \times 10^4$ | 10 | 1+ |
| NOG 2–5 | $8.0 \times 10^4$ | 10 | 2+ |
| NOG 7–2 | $8.5 \times 10^4$ | 20 | 1+ |
| NOG 7–4 | $8.5 \times 10^4$ | 20 | 3+ |
| NOG 7–7 | $8.5 \times 10^4$ | 20 | 3+ |
| NOG 8–8 | $1.0 \times 10^5$ | 20 | 2+ |
| NOG 8–9 | $1.0 \times 10^5$ | 20 | 3+ |
| NOG 8–10 | $1.0 \times 10^5$ | 20 | 2+ |
| EBV-uninfected mice |  |  |  |
| NOG 4–4 | $8.0 \times 10^4$ | NA | – |
| NOG 4–5 | $8.0 \times 10^4$ | NA | – |
| NOG 4–9 | $8.0 \times 10^4$ | NA | – |
| NOG 7–3 | $8.5 \times 10^4$ | NA | – |
| NOG 7–8 | $8.5 \times 10^4$ | NA | – |

EBV, Epstein-Barr virus; NA, Not applicable.

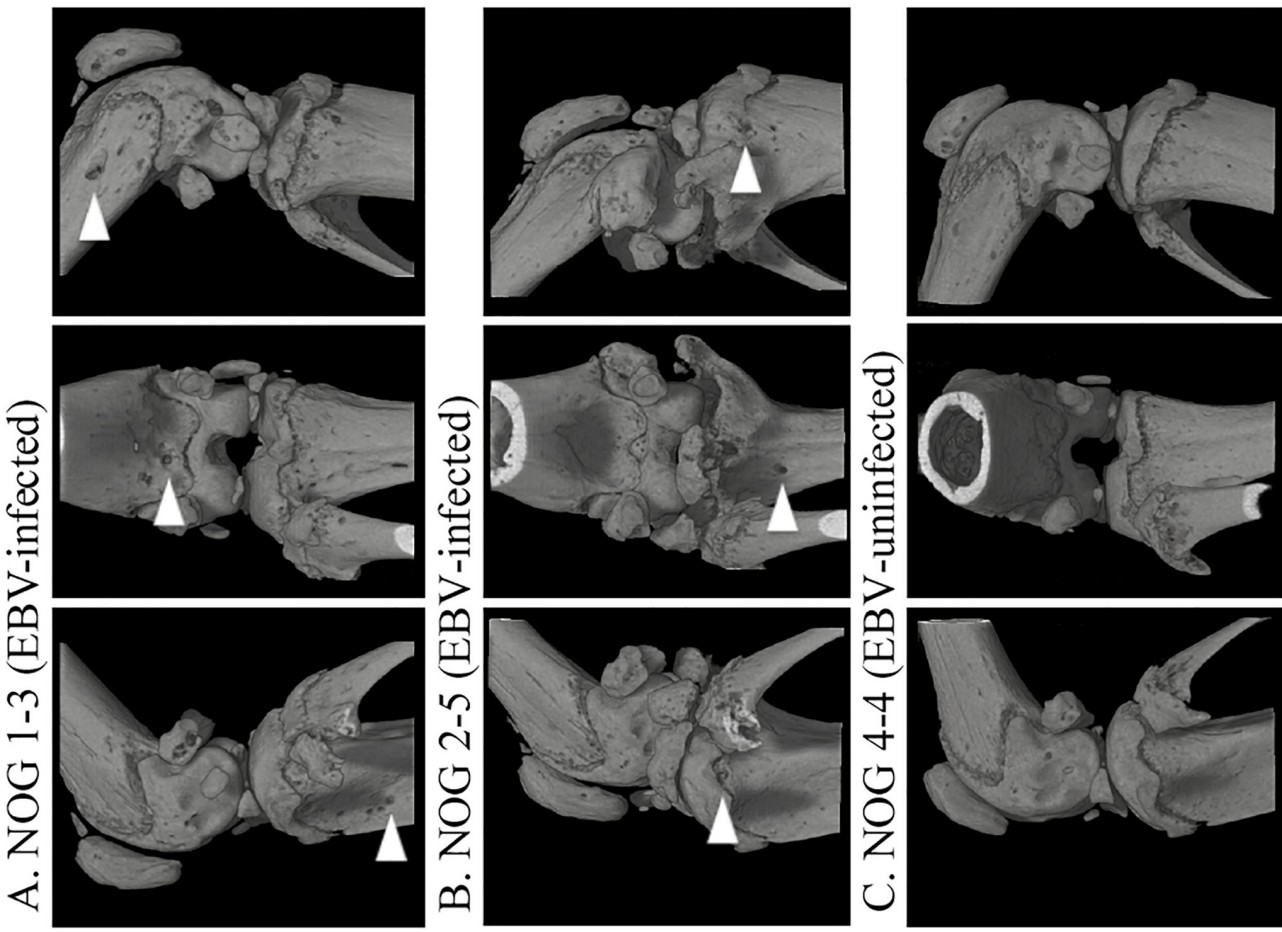

**Fig 2. 3D-CT images of knee joints in EBV-infected and -uninfected hu-NOG mice.** (A) Joint from an EBV-infected mouse. Arrowheads indicate bone erosion at the distal portion of the femur and proximal portion of the tibia. (B) Joint of another EBV-infected mouse. (C) Joint of an EBV-uninfected mouse (control). Republished from [Nagasawa Y, Ikumi N, Nozaki T, Inomata H, Imadome K, et al. Human osteoclasts are mobilized in erosive arthritis of Epstein-Barr virus-infected humanized NOD/Shi-scid/IL2Rγ$^{null}$ mice. 2014 ACR/ARHP Annual Meeting. Abstract number:2340.] under a CC BY license, with permission from [American College of Rheumatology], original copyright [2014].

CBB < 1/3$^{rd}$ of the bone thickness were defined as grades 3+, 2+, and 1+, respectively (Typical knee joint tissue images are shown in S1 Fig). Table 1 shows the severity of bone erosion, the number of transplanted CD34$^{+}$ HSCs, and the dose of inoculated EBV in each mouse. Although variation in the severity of bone erosion was observed among individual EBV-infected mice, correlation between the severity of bone erosion and titer of inoculated EBV was not observed (P = 1.0). Three-dimensional computed tomography (3D-CT) images of knee joints revealed that bone erosive changes resembled RA in EBV-infected mice (n = 2) at locations where histological analysis identified CBB, whereas no such changes were observed in uninfected mice (n = 2). The 3D-CT images of the knee joint in representative mice are shown in Fig 2.

## Properties of multinucleated cells present in bone erosion sites

To determine the characteristics of the multinucleated cells, serial sections of knee joint tissues from EBV-infected (n = 2) and EBV-uninfected mice (n = 2) were stained for TRAP, human cathepsin K, and human MMP-9. Bone erosion lesions were observed in both EBV-infected

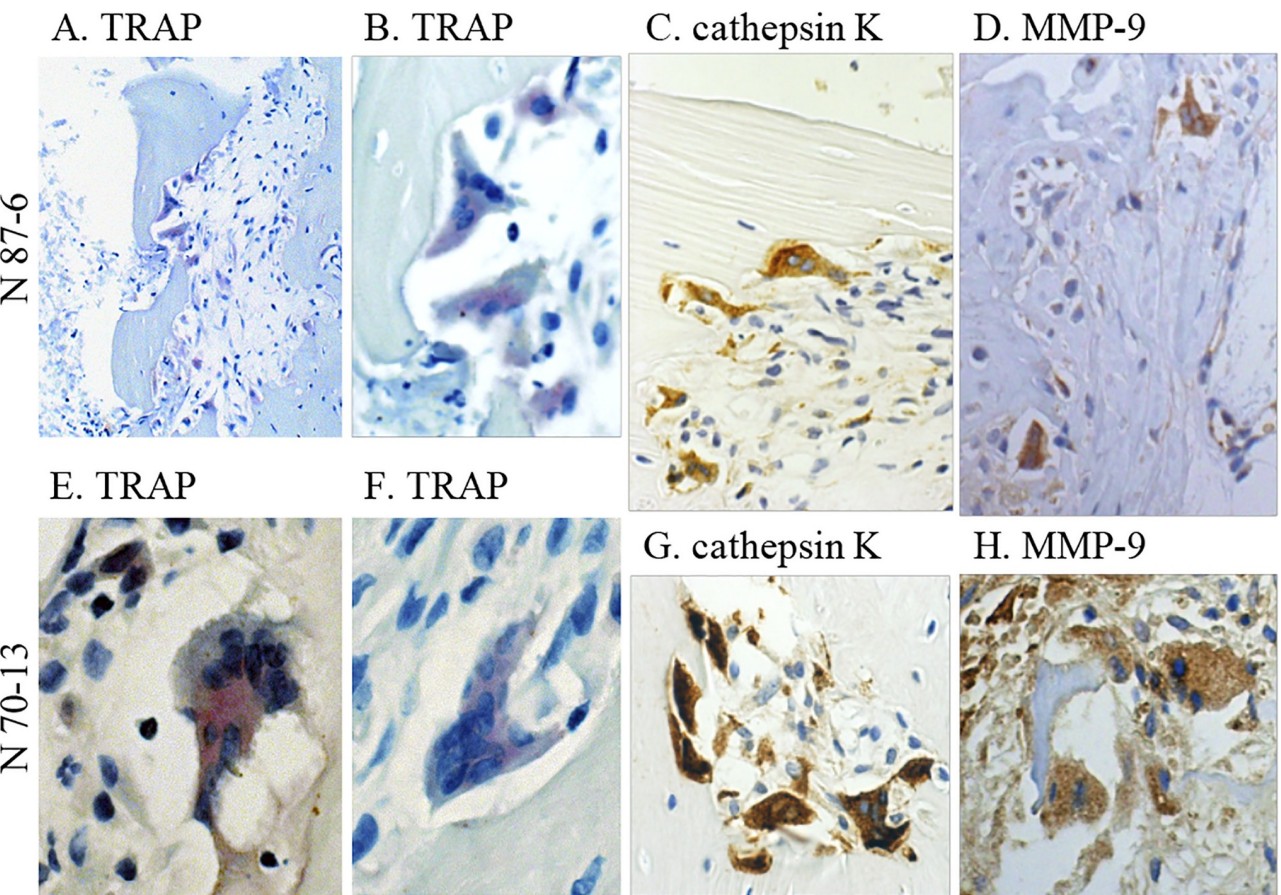

**Fig 3. Histochemistry of the knee joint section of EBV-infected hu-NOG mice.** (A–D) Joint sections from an EBV-infected mouse (N 87–6). (A and B) Joint sections stained for TRAP. TRAP observed in multinucleated cells at the bone erosion site stained red violet. (C) Joint section immunostained for human cathepsin K. Multinucleated cells positive for human cathepsin K stained brown with the anti-human cathepsin K antibody. (D) Joint section immunostained for human MMP-9. Multinucleated cells positive for MMP-9 stained brown with the anti-human MMP-9 antibody (dilution, 1:25). (E–H) Joint sections from another EBV-infected mouse (N 70–13). (E and F) Joint sections stained for TRAP. TRAP-positive multinucleated cells were found in the knee joint tissue. Joint sections immunostained for human cathepsin K (G) and human MMP-9 (dilution, 1:50) (H). These multinucleated cells stained positively with the anti-human cathepsin K antibody and the anti-human MMP-9 antibody. Original magnification, A: 200×, others: 400×. Republished from [Nagasawa Y, Ikumi N, Nozaki T, Inomata H, Imadome K, et al. Human osteoclasts are mobilized in erosive arthritis of Epstein-Barr virus-infected humanized NOD/Shi-scid/IL2Rγ$^{null}$ mice. 2014 ACR/ARHP Annual Meeting. Abstract number:2340.] under a CC BY license, with permission from [American College of Rheumatology], original copyright [2014].

hu-NOG mice, which contained multinucleated osteoclast-like cells. Images of the bone erosion zone in the knee joints of two EBV-infected mice are shown in Fig 3. All multinucleated cells observed in the affected joint tissues showed positive staining for TRAP, and almost all multinucleated cells stained positively with the anti-human cathepsin K antibody and the anti-human MMP-9 antibody. The anti-human cathepsin K and anti-human MMP-9 antibodies used in these experiments do not react with mouse proteins per manufacturer's specifications.

## Differentiation of osteoclasts from the bone marrow cell culture

Human osteoclasts were identified in the bone erosion sites of EBV-infected hu-NOG mice, as mentioned above. However, whether human osteoclasts can develop in HSC-transplanted humanized mice, such as hu-NOG mice, was not known. Therefore, we used a procedure known to induce in vitro osteoclast differentiation on bone marrow cells from long bones of EBV-infected hu-NOG mice (n = 2) and EBV-uninfected hu-NOG mice (n = 2).

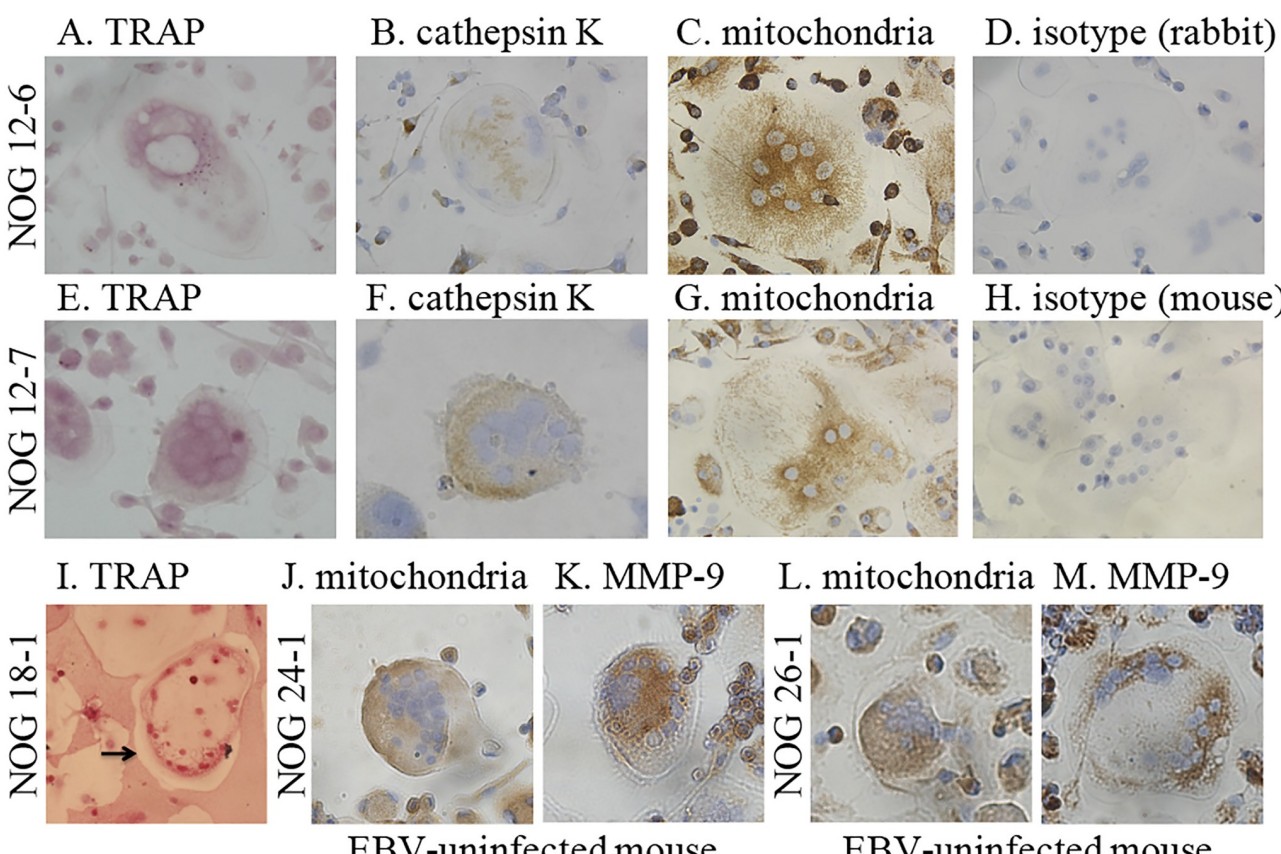

**Fig 4. Cytochemistry of cultured long bone marrow cells from EBV-infected and -uninfected hu-NOG mice in the presence of M-CSF and RANKL differentiation signals.** (A–D) Cultured cells from an EBV-infected mouse (NOG 12–6). (A) Cultured cells stained for TRAP. TRAP in the cultured multinucleated cell stained red violet. (B) Cultured cells immunostained for human cathepsin K. The positive multinucleated cell stained brown with the anti-human cathepsin K antibody (dilution, 1:1800). (C) Cultured cells immunostained for human mitochondrial protein. Multinucleated cell positively stained brown with anti-human mitochondria antibody (dilution, 1:40). (D) Cultured cells immunostained using isotype control rabbit immunoglobulin (dilution, 1:4500). (E–H) Cultured cells from another EBV-infected mouse (NOG 12–7). The multinucleated cells positively stained for TRAP (E), and with anti-human cathepsin K antibody (dilution, 1:600) (F) and anti-human mitochondria antibody (dilution, 1:10) (G). (H) Cultured cells immunostained using isotype control mouse immunoglobulin (dilution, 1:5). (I) Cultured cells from another EBV-infected mouse (NOG 18–1) on a pit formation assay plate. TRAP in the cultured multinucleated cells stained red violet. Calcium phosphate coating around the TRAP-positive multinucleated cell was eliminated (arrow). (J and K) Cultured cells from an EBV-uninfected mouse (NOG 24–1). (J) Cultured cells immunostained for human MMP-9. The multinucleated cell positive for human MMP-9 stained brown with anti-human MMP-9 antibody (dilution, 1:25). (K) Cultured cells immunostained for human mitochondrial protein. The multinucleated cell positive for human mitochondrial protein stained brown with anti-human mitochondria antibody (dilution, 1:10). (L and M) Cultured cells from another EBV-uninfected mouse (NOG 26–1). The multinucleated cells were considered positive when stained with anti-human MMP-9 antibody (dilution, 1:50) (L) and anti-human mitochondria antibody (dilution, 1:40) (M). (original magnification, 400×) Republished from [Nagasawa Y, Ikumi N, Nozaki T, Inomata H, Imadome K, et al. Human osteoclasts are mobilized in erosive arthritis of Epstein-Barr virus-infected humanized NOD/Shi-scid/IL2Ry$^{null}$ mice. 2014 ACR/ARHP Annual Meeting. Abstract number:2340.] under a CC BY license, with permission from [American College of Rheumatology], original copyright [2014].

Multinucleated cells with osteoclast-like morphology were generated after 10–14 days from bone marrow cell culture isolated from EBV-infected mice and EBV-uninfected mice in the presence of human M-CSF and human soluble RANKL. The images of the multinucleated cells cultured are shown in Fig 4. Almost all the multinucleated cells stained positively for TRAP, and several multinucleated cells stained positively with polyclonal antibodies specific for human cathepsin K and human MMP-9. Furthermore, they showed positive staining with a monoclonal antibody specific for the 60 kDa non-glycosylated protein component of human mitochondria.

Next, using the pit formation assay, we investigated whether these human osteoclasts derived from the bone marrow of EBV-infected hu-NOG mice showed the functional characteristics of osteoclasts (n = 3). The results showed that the TRAP-positive multinucleated cells derived from the bone marrow of these EBV-infected mice formed many resorption pits on the surface of plates coated with synthetic bone mimetics (Fig 4).

According to the manufacturer's specifications, the antibodies specific to the human MMP-9 and human mitochondrial protein used in this experiment do not react with mouse proteins. For confirmation, we examined the species specificity of these antibodies. The images of the multinucleated cells cultured from mouse osteoclast progenitor cells are shown in Fig 5. These cultured multinucleated cells were stained positively with antibodies that react with mouse mitochondria and mouse MMP-9 proteins. However, those were not stained with the antibodies specific for the human MMP-9 and human mitochondrial protein.

Furthermore, to determine whether the human osteoclasts developed from bone marrow cells without human RANKL and human M-CSF, the bone marrow cells from pelvic bones of EBV-infected mice (n = 3) were cultured in α-MEM for 8 days. The cultured adherent cells were stained for TRAP and human MMP-9. The images of the cultured representative multinucleated cells are shown in Fig 6. Several multinucleated cells were observed in the cultured bone marrow cells. These multinucleated cells stained positively for TRAP and with the anti-human MMP-9 antibody. In contrast, adherent cells could not be cultured from the bone marrow of EBV-uninfected mice (S2 Fig).

## Discussion

In the present study, we established experimental procedures to reproducibly induce erosive arthritis in hu-NOG mice using EBV inoculation. Using these procedures, we confirmed that EBV indeed induces erosive arthritis, which histologically resembles RA in hu-NOG mice. Furthermore, 3D-CT definitively revealed bone destruction in the knee joints of EBV-infected mice, providing diagnostic imaging evidence similar to that observed in patients with RA. Histological analysis of 10 EBV-infected mice revealed inter-individual variation in the level of bone erosion ranging from 1+ to 3+. Although these data were obtained for other EBV-infected hu-NOG mice under different experimental conditions, we determined the bone mineral density (BMD) of trabecular bone at the distal portion of the femur. Comparison of BMD between mice with severe erosion and those with mild erosion indicated that the former had significantly lower BMD grades than the latter (S3 Fig), suggesting that the severity of bone erosion was affected by the activity level of the bone-resorbing osteoclasts.

### Image of cultured bone marrow cells from EBV-uninfected hu-NOG mice in the absence of differentiation signals in glass-bottom dish

We showed that the multinucleated cells observed in the bone erosion sites of EBV-infected mice were TRAP-positive osteoclasts expressing human cathepsin K and human MMP-9. As osteoclasts are critical effectors of bone destruction, this result strongly suggested that human osteoclasts play a major role in bone erosion of EBV-infected hu-NOG mice. To further confirm this observation, we investigated the presence of human osteoclast progenitor cells in hu-NOG mice and analyzed whether the bone marrow cells of NOG mice possess the potential to differentiate into human osteoclasts. Upon in vitro culture in the presence of human M-CSF and soluble human RANKL, bone marrow cells obtained from EBV-infected mice differentiated into multinucleated cells expressing TRAP, human cathepsin K, and human mitochondrial protein. As these cells showed bone absorption ability in the pit formation assay, we concluded that human mature osteoclasts can differentiate from bone marrow cells of

## A. mouse MMP-9

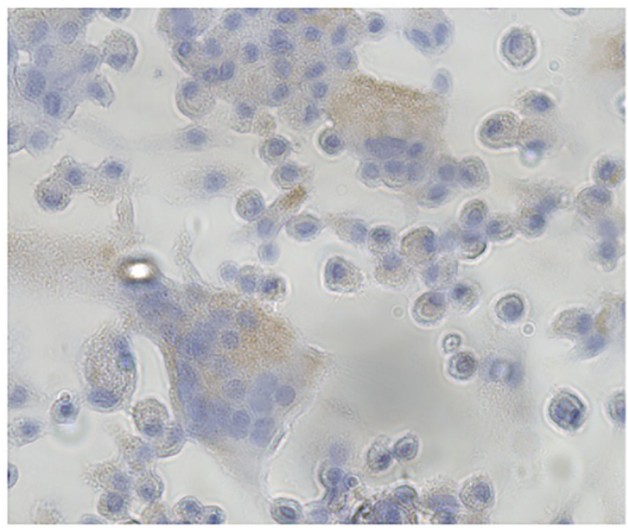

## B. human MMP-9

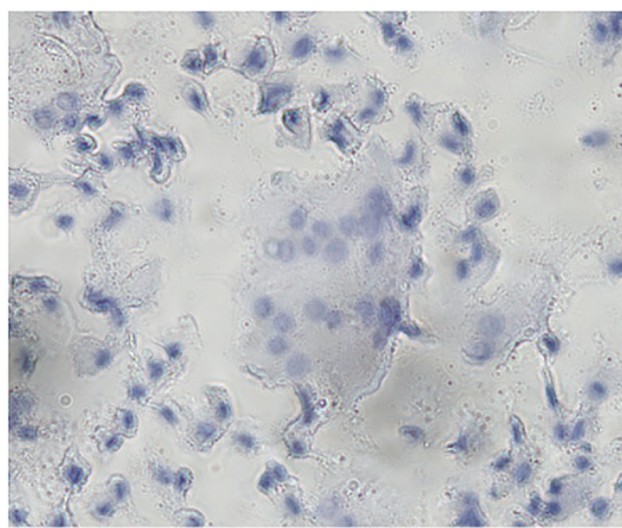

## C. mouse mitochondria

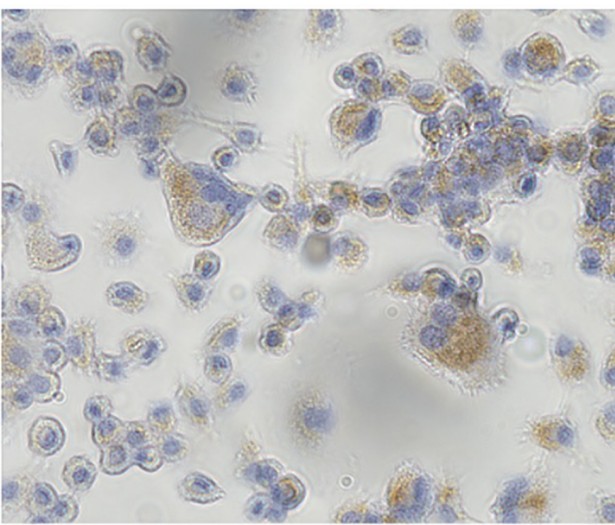

## D. human mitochondria

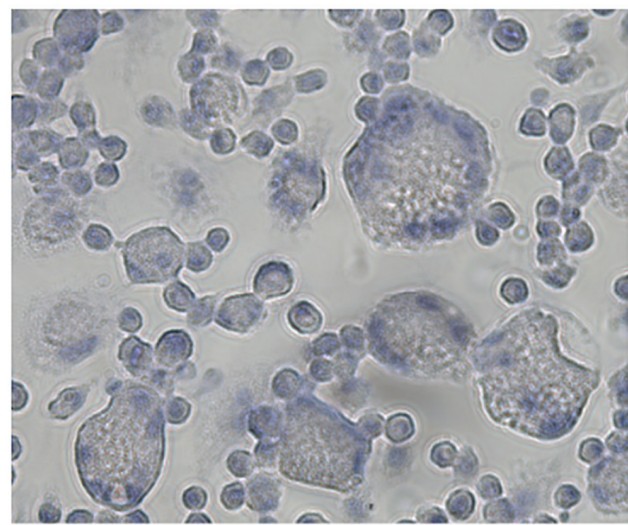

**Fig 5. Cytochemistry of cultured mouse osteoclast progenitor cells in the presence of M-CSF and RANKL differentiation signals.** (A–D) Cultured cells from mouse osteoclast progenitor cells. (A) Cultured cells immunostained with an antibody that reacts with mouse MMP-9. Multinucleated cells positive for MMP-9 stained brown with the anti-MMP-9 antibody (dilution, 1:100). (B) Cultured cells immunostained with an antibody specific to the human MMP-9. Multinucleated cells were not stained (antibody dilution, 1:12.5). (C) Cultured cells immunostained with an antibody that reacts with mouse mitochondrial protein. Multinucleated cells positive for the mouse mitochondrial protein stained brown with the anti-mitochondrial antibody (dilution, 1:360). (D) Cultured cells immunostained with an antibody specific to human mitochondrial protein. Multinucleated cells were not stained (antibody dilution, 1:10). Original magnification, 400×.

EBV-infected hu-NOG mice. To our knowledge, this is the first demonstration of human osteoclast differentiation in humanized mice engrafted with human CD34[+] HSCs. This showed that human osteoclast progenitor cells were present in EBV-infected hu-NOG mice. Based on these results, we concluded that EBV, which do not infect mouse cells, infected human cells differentiated from HSCs transplanted in NOG mice, and EBV infections induced human osteoclast differentiation from human osteoclast progenitor cells and resulted in bone erosion.

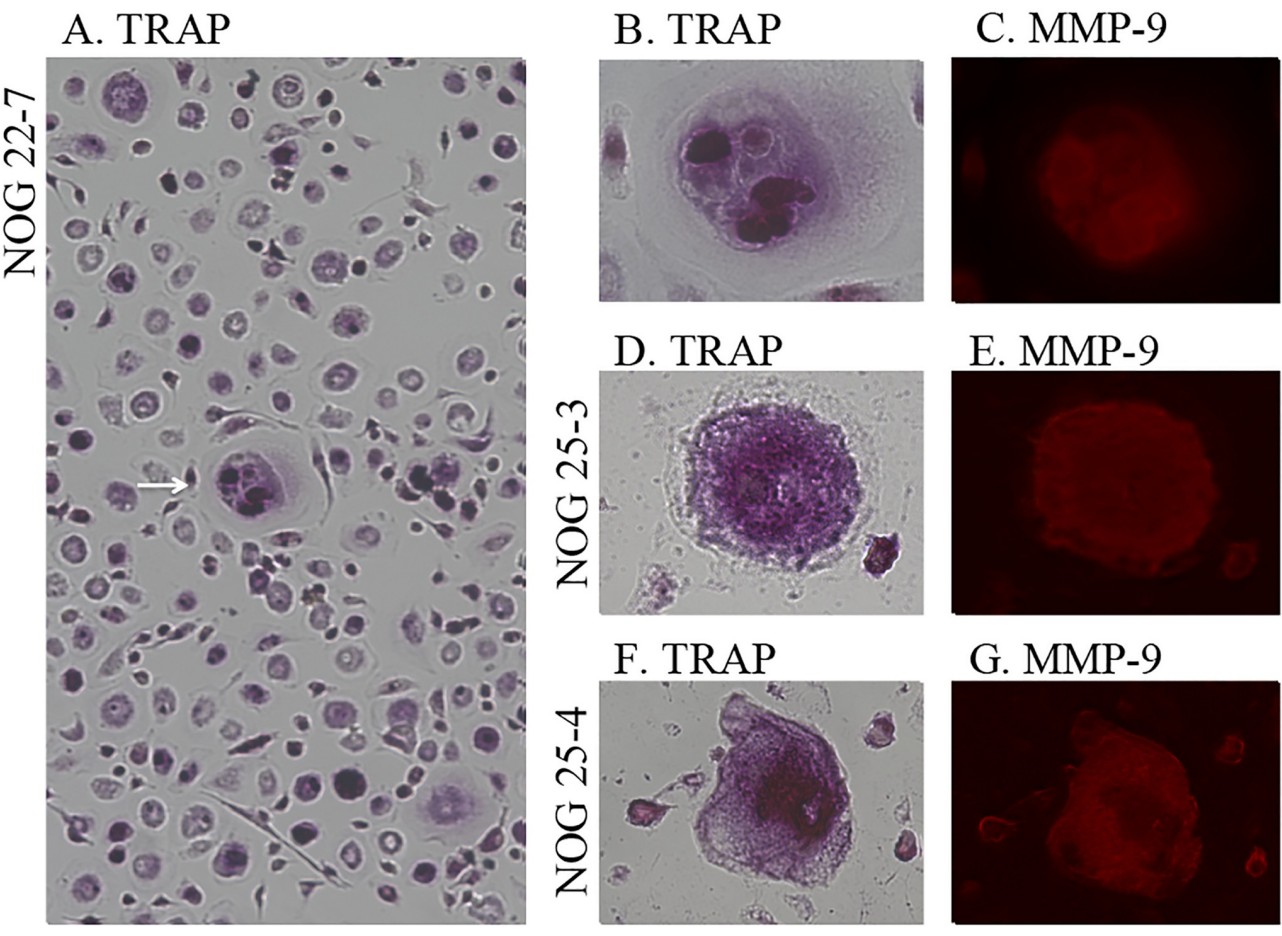

**Fig 6. Cytochemistry of cultured pelvic bone marrow cells from EBV-infected hu-NOG mice in the absence of differentiation signals.** (A–C) Cultured cells from an EBV-infected mouse (NOG 22–7) in glass-bottom dishes. (A and B) Cultured cells stained for TRAP. (A) Multinucleated cells were confirmed in the bone marrow (arrow). (B) The same cultured cell stained for TRAP. TRAP in the cultured multinucleated cell stained red violet. (C) The same cultured cell immunostained for human MMP-9. The TRAP-positive multinucleated cell stained red with the anti-human MMP-9 antibody. (D and E) Cultured cells from another mouse (NOG 25–3). (D) Cultured cell stained for TRAP. The multinucleated cell stained positive for TRAP. (E) The same cultured cell immunostained for human MMP-9. Multinucleated cell positive for anti-human MMP-9 antibody were stained. (F and G) Cultured cells from another mouse (NOG 25–4). (F) Cultured cell stained for TRAP. (G) The same cultured cell immunostained for human MMP-9. Original magnification, A: 100×, others: 400×.

Upon in vitro culture in the presence of human M-CSF and soluble human RANKL, bone marrow cells obtained from EBV-uninfected mice differentiated into multinucleated cells expressing human cathepsin K, human MMP-9, and human mitochondrial protein. This result indicates that similar human osteoclast progenitor cells are present in the bone morrow cells of EBV-uninfected mice. In the in vitro bone marrow cell culture of EBV-infected hu-NOG mice lacking the human M-CSF and soluble human RANKL signaling molecules, the multinucleated cells expressed TRAP and human MMP-9. In contrast, adherent cells, which tended to develop into osteoclasts in in vitro cell culture, could not be cultured from the bone marrow cells of EBV-uninfected mice. This indicated that the bone marrow of EBV-infected hu-NOG mice provided an environment in which human osteoclast progenitor cells differentiated into human osteoclasts and induced their excessive differentiation to human osteoclasts and this aberrant activation resulted in bone erosion in hu-NOG mice.

Osteoclast differentiation from their progenitors requires M-CSF binding to its receptor CSF1R (CD115) and RANKL binding to RANK. In normal bone remodeling, osteoblasts provide RANKL and M-CSF for osteoclastogenesis, and bone homeostasis is maintained by the balance between bone-resorbing osteoclasts and bone-synthesizing osteoblasts [34]. Therefore, excessive M-CSF and/or RANKL expression possibly results in excessive osteoclastogenesis. M-CSF expression in RA has been reported to increase in the synovial tissue [35]. Furthermore, some reports state that synovial fibroblasts and/or activated T lymphocytes produce RANKL, which triggers aberrant differentiation and activation of osteoclasts, causing bone destruction [36,37].

What is the mechanism underlying human osteoclast differentiation and activation in EBV-infected hu-NOG mice? Regarding CSF1R, reports have shown that along with M-CSF, IL-34 also functions as an agonist. In addition, mouse M-CSF does not react with human CSF1R, although the amino acid sequences of mouse IL-34 is highly homologous to that of human IL-34 [38]. Therefore, human osteoclast progenitor cells possibly receive the differentiation signal from human M-CSF, human IL-34, and/or mouse IL-34 in EBV-infected hu-NOG mice. The tissue distribution of M-CSF and IL-34 differs. Reports have shown that osteoclasts in the osseous tissue are differentiated and maintained by M-CSF, whereas those in the spleen are maintained by M-CSF and/or IL-34 [39]. Thus, the mechanism underlying osteoclast differentiation involving CSF1R, which exists in the human osteoclast progenitor cells of EBV-infected hu-NOG mice, might be complex. In the absence of evidence showing that mouse RANKL binds to human RANK, it appears unlikely that cells of mouse origin, including fibroblasts and stromal cells, may participate in human osteoclast differentiation via mouse RANKL-human RANK interaction.

EBV primarily infects human B cells and transforms them into lymphoblastoid cells of the activated B-cell phenotype [23]. EBV-specific T-cell responses have been demonstrated in EBV-infected hu-NOG mice [24]. EBV-infected lymphoblastoid cells and/or activated T cells responding to the virus might have triggered the aberrant differentiation and activation of human osteoclasts in hu-NOG mice. In our previous study, we identified numerous EBV-infected cells, as well as both CD4[+] and CD8[+] human T cells, in the edematous bone marrow adjacent to the affected joints, whereas only a few EBV-infected cells were present in the synovial tissue of these joints [25]. Hence, we suggested that in the bone marrow, EBV-infected cells and/or responding human T cells may induce differentiation and activation of osteoclasts, leading to bone erosion. Further investigations are required to elucidate the mechanism of differentiation and excessive activation of human osteoclasts in EBV-infected hu-NOG mice.

The present results suggest that EBV infections promote human osteoclast differentiation in the present mouse model, resulting in erosive arthritis, and the EBV-infected hu-NOG mice have the reproducibility of erosive arthritis resembling RA, which can be established in a mouse model. However, the mechanisms underlying human osteoclastogenesis remain unclear. Further studies are required to determine the sources of human osteoclast differentiation signals including RANKL and M-CSF. Furthermore, this model can help confirm the therapeutic effects of anti-human RANKL antibody or cathepsin K inhibitor. However, a mouse model displaying activated human osteoclasts is extremely valuable and is potentially applicable in studies on the association between EBV infections and RA and with human bone metabolism using mouse models such as those of osteoporosis.

## Supporting information

**S1 Fig. Histochemistry of the knee joint section of representative EBV-infected and -uninfected hu-NOG mice.** Knee joint sections. (A) Severe bone erosion, defined as grade 3+, in an

EBV-infected mouse. (B) Mild bone erosion, defined as grade 2+, in an EBV-infected mouse. (C) Slight bone erosion, defined as grade 1+, in an EBV-infected mouse. (D) Lack of bone erosion in an uninfected mouse. Original magnification, 100×.
(TIF)

**S2 Fig. Cultured bone marrow cells from EBV-uninfected hu-NOG mice in the absence of differentiation signals in glass-bottom dish.** Adherent cells with a tendency to develop into osteoclasts in vitro could not be cultured. Original magnification, 40×.
(TIFF)

**S3 Fig. Comparison of trabecular bone at the distal portion of the femur between a severe erosion group and a mild erosion group.** Bone mineral density was compared between EBV-infected mice having 3+ or 2+ bone erosion (n = 5) and EBV-infected mice having 1+ bone erosion (n = 5) or EBV-uninfected mice having no bone erosion (n = 5). Severe bone erosion group had significantly lower BMD grades than those of mild erosion group. Values represent mean ± standard deviation. Statistical analysis was performed using student's t-test (*P value = 0.0439).
(TIFF)

## Acknowledgments

We thank Ms. Eiko Ishizuka for excellent technical support.

## Author Contributions

**Conceptualization:** Masami Takei.

**Data curation:** Yosuke Nagasawa.

**Formal analysis:** Yosuke Nagasawa, Masami Takei, Mitsuhiro Iwata, Yasuko Nagatsuka, Kenichi Imai, Ken-Ichi Imadome, Shigeyoshi Fujiwara, Noboru Kitamura.

**Funding acquisition:** Masami Takei.

**Investigation:** Yosuke Nagasawa, Hiroshi Tsuzuki.

**Methodology:** Masami Takei.

**Project administration:** Masami Takei.

**Supervision:** Yasuko Nagatsuka, Noboru Kitamura.

**Validation:** Masami Takei.

**Writing – original draft:** Yosuke Nagasawa, Mitsuhiro Iwata.

**Writing – review & editing:** Masami Takei, Mitsuhiro Iwata, Yasuko Nagatsuka, Kenichi Imai, Ken-Ichi Imadome, Shigeyoshi Fujiwara, Noboru Kitamura.

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
