## [Decision Letter · Decision Letter 0]

11 Sep 2020

PONE-D-20-19201

Human osteoclastogenesis in Epstein-Barr virus-induced erosive arthritis in humanized
NOD/Shi-scid/IL-2Rγnull mice

PLOS ONE

Dear Dr. Takei,

Thank you for submitting your manuscript to PLOS ONE.Your manuscripts are reviewed by
two experts in different fields.  I also read your manuscript quickly. We all agree
that this is an interesting and potentially important work.  However, they also
identified several significant concerns that require further attention. As indicated
in the appended comments, a number of specific points have been raised that have to
be resolved. Based on the combined assessments of the reviewers, we would be willing
to provide you with an opportunity to respond to these issues in a suitably revised
version of the manuscript.  After careful consideration, we feel that your
manuscript has merit and will be reconsidered for publication in PLoS One after
major revision.

Please submit your revised manuscript by Oct 26 2020 11:59PM. If you will need more
time than this to complete your revisions, please reply to this message or contact
the journal office at plosone@plos.org. When
you're ready to submit your revision, log on to https://www.editorialmanager.com/pone/ and select the 'Submissions
Needing Revision' folder to locate your manuscript file.

If you would like to make changes to your financial disclosure, please include your
updated statement in your cover letter. Guidelines for resubmitting your figure
files are available below the reviewer comments at the end of this letter.

We look forward to receiving your revised manuscript.

Kind regards,

Luwen Zhang

Academic Editor

PLOS ONE

Journal Requirements:

2.In your submission, you state that your work was presented in abstract form at the
2014 american college of rheumatology annual meeting.

Please clarify whether this abstract is under copyright, and whether the copyright
owner is the authors or the college of rheumatology. If the copyright is not held by
the authors, permission from the original copyright holder to publish under a CC BY
license needs to be obtained and the following should be completed:

Please provide proof that the owner of the content (a) has given you written
permission to use it, and (b) has approved of the CC BY license being applied to
their content. You may have the following form completed by the owner as proof:
https://journals.plos.org/plosone/s/file?id=7c09/content-permission-form.pdf.
Alternatively, you may electronically request permissions through from the copyright
holder and send us proof of approval, as long as the approval clearly shows that the
owner has approved of the CC BY license being applied to their content. Please see
https://journals.plos.org/plosone/s/licenses-and-copyright for more
information.

3. Please provide the dilutions of all antibodies used in your study.

4. Please include further information regarding your animal research, per our
guidelines (http://journals.plos.org/plosone/s/submission-guidelines#loc-animal-research).
Specifically, please provide details regarding:

 1) Animal health monitoring, including:

     -frequency of monitoring,

     -monitoring criteria, and

     -any efforts made to reduce suffering and distress, such as administering

      analgesics.

 2) any mortality that occurred outside of planned euthanasia

 3) The total number of mice.

Reviewers' comments:

Reviewer's Responses to Questions

**Comments to the Author**

1. Is the manuscript technically sound, and do the data support the conclusions?

Reviewer #1: Partly

Reviewer #2: Yes

2. Has the statistical analysis been performed
appropriately and rigorously? 

Reviewer #1: No

Reviewer #2: Yes

3. Have the authors made all data underlying the
findings in their manuscript fully available?

Reviewer #1: Yes

Reviewer #2: Yes

4. Is the manuscript presented in an intelligible
fashion and written in standard English?

Reviewer #1: Yes

Reviewer #2: Yes

5. Review Comments to the Author

Reviewer #1: Human osteoclastogenesis in Epstein-Barr virus-induced erosive arthritis
in humanized NOD/Shi-scid/IL-2Ryneull mice by Nagasawa et al describes a humanized
mouse model in which erosive arthritis during EBV infections can be studied. Their
results suggest that human osteoclasts induce erosive arthritis during EBV
infection.

This is exciting and significant work but proper controls need to be shown. Without
the necessary controls this review can’t properly evaluate the quality of the data.
These controls should be added in addition to increasing the number of
representative pictures and sample sizes as described below.

Results section. Page 18. Outline the and describe the experiment broadly before
jumping into the specific results to make it easier for the reader to follow.

Figure 1. The legend needs to be more descriptive. Are these averages of the 10
infected samples?

Error bars need to be included and statistics should be provided. The data for the
uninfected samples needs to be included.

Table 1. Why were different numbers of CD34+ cells used for different sets? What is
the rational for different amounts?

Figure 2. Images are only shown for 1 infected and 1 uninfected mouse. More
representative pictures should be added. I would like to see pictures representing
the different severity of bone erosion described in Table 1. (ie samples from 1, 2,
and 3+ in addition to the negative control).

Figure 3. Only 2 mice from EBV infected or uninfected where examined. 10 mice were
infected – including more samples would increase the significance. Only one picture
is shown for TRAP, Cathepsin and MMP-9 staining. The authors state these antibodies
do not react with the mouse proteins but the negative controls are necessary and
need to be shown.

Figure 4. An n=2 is low and only single pictures are shown. Negative controls are
necessary and should be shown.

Figure 5. n=3 but only one sample is shown.

Supplemental figure doesn’t have a legend.

Reviewer #2: The article “Human osteoclastogenesis in Epstein-Barr virus-induced
erosive arthritis in humanized NOD/Shi-scid/IL-2Rgnull mice” is an interesting
original study that starts to elucidate the mechanisms of EBV-induced erosive
arthritis through a hu-NOG model. The study aims are addressed by its methodology
and the three-dimensional computed tomography was a visually appealing tool to
demonstrate bone erosive changes. The presentation of results is linear and the data
is properly discussed, supporting the authors conclusions. Although this is true,
there are some unanswered questions and revisions that will make this study more
impactful.

Major Comments

The authors do not show in vitro osteoclast differentiation of bone marrow cells
(cultured with human RANKL and M-CSF) from EBV-uninfected NOG (hu-NOG only).

Even though they say that EBV infection did not increase the number of osteoclast
progenitors (in a data not shown - sentence line 440-446) they do not provide
evidence that EBV-uninfected hu-NOG have human osteoclast progenitor cells in their
bone marrow, or that these progenitors are able to differentiate in osteoclast in
vitro as for the EBV-infected hu-NOG.

I believe that if the authors provide the data showing in vitro osteoclast
differentiation of bone marrow cells from EBV-uninfected hu-NOG, cultured in the
presence of human RANKL and human M-CSF it would demonstrate potential for
differentiation as well as equal progenitor

Typically in normal bone remodeling there is a balance between osteoblasts and
osteoclasts, with the former being responsible for inducing the differentiation of
the latter through the production of RANK-L and M-CSF. It’s true that in pathogenic
settings the osteoblasts are not the main sources of these cytokines, but they are
still important for conferring some osteoprotection. Although the authors bring this
to their discussion, there is no information on osteoblasts in their data. It would
be informative to understand the ratio of osteoblast to osteoclasts in hu-NOG and
after infection with EBV.

Minor comments

In the methods section the authors list CD19+ as one of the monitored percentages of
human cells in the blood of hu-NOG. In Fig. 1 the authors do not show CD19+ numbers,
although one can assume by subtracting CD4+ and CD8+ from total Human CD45+, it
would be easier to observe how the frequency of these cells stay through the
experiment course if a CD19+ curve was displayed.

Fig. 5 - it appears that the cells stained for TRAP (in Fig. 5A and B) were also
stained for hematoxylin, correct? If that is true, since this was performed in a
glass bottom dish, the statement in line 219 of the methods section “After washing,
only the plates of serial sections were stained in hematoxylin (cultured cells on
the chamber slides, pit formation assay plates, and glass bottom dishes were not
stained” is incorrect.

Line 645 - Possible typo/missing verb “be cultured”

Have the authors considered staining for collagen (with picrosirius red staining or
by using polarized light microscopy) to assess cathepsin K effects in the joints of
EBV-infected hu-NOG.

Aside RANKL and M-CSF there are a number of inflammatory cytokines that are usually
present in inflamed joints that impact bone remodeling within that microenvironment.
Given that the bone marrow cells of EBV-infected hu-NOG when cultured in vitro
(without human RANKL and M-CSF cytokines) resulted in osteoclast differentiation and
the authors had previously shown bone marrow edema in EBV-infected mice, one could
assume that pro-inflammatory cytokines have an important role in this process.
Specially since T cells do not survive for long in culture, even with proper
stimuli, cells from the monocytic/macrophage lineage could be the sources of this
cytokines in this case. Have the authors investigated IL-1 (alpha and or betta) and
TNF-a (that have been previously implicated in osteoclastogenesis), in the bone
marrow of EBV-infected hu-NOG?

6. PLOS authors have the option to publish the peer
review history of their article (what does this mean?). If published, this will
include your full peer review and any attached files.

If you choose “no”, your identity will remain anonymous but your review may still be
made public.

**Do you want your identity to be public for this peer review?** For
information about this choice, including consent withdrawal, please see our
Privacy Policy.

Reviewer #1: No

Reviewer #2: **Yes: **Thiago Alves da Costa

---

## [Author Response · Author response to Decision Letter 0]

20 Jan 2021

Reviewer #1: Human osteoclastogenesis in Epstein-Barr virus-induced erosive arthritis
in humanized NOD/Shi-scid/IL-2Ryneull mice by Nagasawa et al describes a humanized
mouse model in which erosive arthritis during EBV infections can be studied. Their
results suggest that human osteoclasts induce erosive arthritis during EBV
infection.

This is exciting and significant work but proper controls need to be shown. Without
the necessary controls this review can’t properly evaluate the quality of the data.
These controls should be added in addition to increasing the number of
representative pictures and sample sizes as described below.

Response: Thank you for your constructive criticism and helpful suggestions. We have
evaluated our antibodies on murine osteoclasts and confirmed that they do not react
with mouse osteoclasts. This indicates that those osteoclasts detected in the
arthritis lesions of EBV-infected humanized mice are truly of human origin. In
addition, we have shown data from additional mice that support our conclusion. We
believe our revised manuscript provides more concrete evidence of human osteoclast
differentiation in EBV-infected humanized mice and their involvement in RA-like
erosive arthritis. 

Point by point response

Results section. Page 18. Outline the and describe the experiment broadly before
jumping into the specific results to make it easier for the reader to follow.

Response: Our previous work demonstrated consistent and rapid increase in the number
of peripheral blood CD8+ T cells and reversal of CD4/CD8 ratio in EBV-infected
humanized NOG mice, which constitute the flow-cytometric hallmarks of successful EBV
infection. This study starts with reproduction of these results and then goes on to
further characterize bone erosion induced by EBV. To make this outline clearer, we
have added the following sentence at the beginning of the Results.

 “In this study, we initially confirmed our previous findings of rapid increase in
the number of CD8+ cells and the reversal of CD4/CD8 ratio in the peripheral blood
of humanized mice following EBV infection.” 

Figure 1. The legend needs to be more descriptive. Are these averages of the 10
infected samples?

Error bars need to be included and statistics should be provided. The data for the
uninfected samples needs to be included.

Response: We have revised Figure 1 legend and described the experiment in more
detail. The data shown in this figure are not the average of 10 mice but those from
a representative mouse. In a previous publication, we demonstrated rapid increase in
the number of peripheral blood CD8+ T cells and the reversal of CD4/CD8 ratio as
very consistent responses following EBV infection of humanized mice (Yajima et al, J
Infect Dis 198:673–682, 2008). Figure 1 depicts the outline of changes in the
profile of human lymphocytes following EBV infection, and the indicators to confirm
that our humanized NOG mice are certainly infected with EBV. The data from
uninfected mice were shown in a previous publication (Yajima et al, J Infect Dis
198:673–682, 2008), which indicated that while the percentage of CD3+ T cells among
peripheral human CD45+ leukocytes increased gradually, that of CD19+ B cells
decreased. The CD4/CD8 ratio remained >1 always. 

The revised Figure 1 legend: “Fig 1. Time course of human lymphocyte reconstitution
in the peripheral blood of an EBV-infected hu-NOG mouse. The percentage of human
CD45+, CD4+, CD8+, or CD19+ cells among the peripheral blood mononuclear cells was
measured weekly, after inoculation with EBV. Upon EBV infection, the percentage of
human CD8+ T cells increased rapidly and the ratio of CD4+ cells to CD8+ cells
decreased to below 1. Data from a representative mouse are shown.” 

Table 1. Why were different numbers of CD34+ cells used for different sets? What is
the rational for different amounts? 

Response: In our previous publication, we showed that transplantation of 1 × 104 to
1.2 × 105 cells/mouse of CD34+ HSCs resulted in consistent reconstitution of human
lymphocytes and monocyte/macrophages. In this study, the number of HSCs/mouse was
varied within this range, depending on the number of available mice and available
HSCs. The efficiency of immune reconstitution was not affected by this variation of
HSCs input. 

Figure 2. Images are only shown for 1 infected and 1 uninfected mouse. More
representative pictures should be added. I would like to see pictures representing
the different severity of bone erosion described in Table 1. (ie samples from 1, 2,
and 3+ in addition to the negative control).

Response: We have added pictures of another infected mouse and from other angles in
the revised manuscript. We could afford to use only 2 infected mice and 2 uninfected
mice in three-dimensional computed tomography, because other mice were examined
through other analyses. This figure, however, clearly shows bone destruction lesions
similar to those revealed in patients with RA using the same diagnostic method.
Typical examples of bone erosion 1+, 2+, and 3+ in addition to the negative control
in the knee joint tissue are shown in Supplementary Figure S1.

Figure 3. Only 2 mice from EBV infected or uninfected where examined. 10 mice were
infected – including more samples would increase the significance. Only one picture
is shown for TRAP, Cathepsin and MMP-9 staining. The authors state these antibodies
do not react with the mouse proteins but the negative controls are necessary and
need to be shown.

Response: In the revised Figure 3, we show immunohistochemical data from an
additional mouse that shows the same results. We have obtained mouse osteoclasts by
culturing mouse osteoclast progenitor cells with mouse M-CSF and mouse RANKL. These
mouse osteoclasts did not react with the antibodies against human MMP-9 and
mitochondria, while they were stained positively by antibodies specific for murine
MMP-9 and mitochondria, indicating that those multinuclear cells found in
EBV-infected humanized mice are truly of human origin. The results are shown in the
revised Figure 5. Regarding staining with anti-cathepsin K antibody, additional
experiments could not be planned because the antibody had been discontinued. 

Figure 4. An n=2 is low and only single pictures are shown. Negative controls are
necessary and should be shown.

Response: We have shown results from 3 additional mice in the revised Figure 4. We
could obtain consistent results from these two mice, and therefore, we think the
results are reliable. As stated in the reply to the previous comment, we have shown
that our antibodies did not react with murine osteoclasts and the results are shown
in Figure 5. 

Figure 5. n=3 but only one sample is shown.

Response: In the revised Figure 6, we present data from 2 additional mice that show
the same result.

Supplemental figure doesn’t have a legend.

Response: We have revised the Figure S2 legend and described the experiment in more
detail.

Reviewer #2: The article “Human osteoclastogenesis in Epstein-Barr virus-induced
erosive arthritis in humanized NOD/Shi-scid/IL-2Rgnull mice” is an interesting
original study that starts to elucidate the mechanisms of EBV-induced erosive
arthritis through a hu-NOG model. The study aims are addressed by its methodology
and the three-dimensional computed tomography was a visually appealing tool to
demonstrate bone erosive changes. The presentation of results is linear and the data
is properly discussed, supporting the authors conclusions. Although this is true,
there are some unanswered questions and revisions that will make this study more
impactful.

Response: We thank you for your favorable comments and helpful suggestions. We have
included the data for reconstruction of human CD19+ B cells in Figure 1. We have
also modified the Materials and methods to correct our inconsistent statements
concerning histochemistry. We believe that the manuscript has been sufficiently
improved, and the revised manuscript supports our conclusion adequately. 

Point by point response

Major Comments

The authors do not show in vitro osteoclast differentiation of bone marrow cells
(cultured with human RANKL and M-CSF) from EBV-uninfected NOG (hu-NOG only).

Even though they say that EBV infection did not increase the number of osteoclast
progenitors (in a data not shown - sentence line 440-446) they do not provide
evidence that EBV-uninfected hu-NOG have human osteoclast progenitor cells in their
bone marrow, or that these progenitors are able to differentiate in osteoclast in
vitro as for the EBV-infected hu-NOG.

I believe that if the authors provide the data showing in vitro osteoclast
differentiation of bone marrow cells from EBV-uninfected hu-NOG, cultured in the
presence of human RANKL and human M-CSF it would demonstrate potential for
differentiation as well as equal progenitor

Response: In response to reviewer’s comment, we cultured bone marrow cells of
EBV-uninfected hu-NOG mice in the presence of human M-CSF and human RANKL
differentiation signals. As the reviewer pointed out, EBV-uninfected hu-NOG mice
have human osteoclast progenitor cells in their bone marrow.

Typically in normal bone remodeling there is a balance between osteoblasts and
osteoclasts, with the former being responsible for inducing the differentiation of
the latter through the production of RANK-L and M-CSF. It’s true that in pathogenic
settings the osteoblasts are not the main sources of these cytokines, but they are
still important for conferring some osteoprotection. Although the authors bring this
to their discussion, there is no information on osteoblasts in their data. It would
be informative to understand the ratio of osteoblast to osteoclasts in hu-NOG and
after infection with EBV.

Response: We agree with the reviewer that the osteoblast/osteoclast ratio is
important for bone homeostasis, but we have not yet examined the ratio in our
EBV-infected mice. However, we believe our data are sufficient to claim that EBV
infection of the humanized mice resulted in the differentiation of human osteoclasts
that are most likely responsible for bone erosion found in these mice. 

Minor comments

In the methods section the authors list CD19+ as one of the monitored percentages of
human cells in the blood of hu-NOG. In Fig. 1 the authors do not show CD19+ numbers,
although one can assume by subtracting CD4+ and CD8+ from total Human CD45+, it
would be easier to observe how the frequency of these cells stay through the
experiment course if a CD19+ curve was displayed.

Response: We have shown the CD19+ curve in the revised Figure 1. 

Fig. 5 - it appears that the cells stained for TRAP (in Fig. 5A and B) were also
stained for hematoxylin, correct? If that is true, since this was performed in a
glass bottom dish, the statement in line 219 of the methods section “After washing,
only the plates of serial sections were stained in hematoxylin (cultured cells on
the chamber slides, pit formation assay plates, and glass bottom dishes were not
stained” is incorrect.

Response: We have revised the Methods section and restated that “After washing,
serial sections in the plates and cultured bone marrow cells of pelvic bones in
glass bottom dishes were stained with hematoxylin (cultured cells on the chamber
slides and pit formation assay plates were not stained).”

Line 645 - Possible typo/missing verb “be cultured”

Response: We have revised as suggested.

Have the authors considered staining for collagen (with picrosirius red staining or
by using polarized light microscopy) to assess cathepsin K effects in the joints of
EBV-infected hu-NOG.

Response: We have not stained for collagen in our mouse samples, although we expect
to see degradation. We plan to perform further immunohistochemical characterization
of our mouse arthritis model during the next stage of our research. 

Aside RANKL and M-CSF there are a number of inflammatory cytokines that are usually
present in inflamed joints that impact bone remodeling within that microenvironment.
Given that the bone marrow cells of EBV-infected hu-NOG when cultured in vitro
(without human RANKL and M-CSF cytokines) resulted in osteoclast differentiation and
the authors had previously shown bone marrow edema in EBV-infected mice, one could
assume that pro-inflammatory cytokines have an important role in this process.
Specially since T cells do not survive for long in culture, even with proper
stimuli, cells from the monocytic/macrophage lineage could be the sources of this
cytokines in this case. Have the authors investigated IL-1 (alpha and or betta) and
TNF-a (that have been previously implicated in osteoclastogenesis), in the bone
marrow of EBV-infected hu-NOG?

Response: We have recently initiated the analysis on the mechanism of osteoclast
differentiation induced by EBV and believe it is necessary to examine the role of
human inflammatory cytokines, including IL-1 and TNF-α. We have not performed the
experiments yet.

---

## [Decision Letter · Decision Letter 1]

15 Feb 2021

PONE-D-20-19201R1

Human osteoclastogenesis in Epstein-Barr virus-induced erosive arthritis in humanized
NOD/Shi-scid/IL-2Rγnull mice

PLOS ONE

Dear Dr. Takei,

Thank you for submitting your manuscript to PLOS ONE. After careful consideration, we
feel that it has merit but does not fully meet PLOS ONE’s publication criteria as it
currently stands. Reviewers still have some concerns. Therefore, we invite you to
submit a revised version of the manuscript that addresses the points raised during
the reviewers.

Please submit your revised manuscript by Apr 01 2021 11:59PM. If you will need more
time than this to complete your revisions, please reply to this message or contact
the journal office at plosone@plos.org. When
you're ready to submit your revision, log on to https://www.editorialmanager.com/pone/ and select the 'Submissions
Needing Revision' folder to locate your manuscript file.

If you would like to make changes to your financial disclosure, please include your
updated statement in your cover letter. Guidelines for resubmitting your figure
files are available below the reviewer comments at the end of this letter.

We look forward to receiving your revised manuscript.

Kind regards,

Luwen Zhang

Academic Editor

PLOS ONE

Reviewers' comments:

Reviewer's Responses to Questions

**Comments to the Author**

1. If the authors have adequately addressed your comments raised in a previous round
of review and you feel that this manuscript is now acceptable for publication, you
may indicate that here to bypass the “Comments to the Author” section, enter your
conflict of interest statement in the “Confidential to Editor” section, and submit
your "Accept" recommendation.

Reviewer #1: (No Response)

Reviewer #2: All comments have been addressed

2. Is the manuscript technically sound, and do the data
support the conclusions?

Reviewer #1: Yes

Reviewer #2: Yes

3. Has the statistical analysis been performed
appropriately and rigorously? 

Reviewer #1: Yes

Reviewer #2: Yes

4. Have the authors made all data underlying the
findings in their manuscript fully available?

Reviewer #1: Yes

Reviewer #2: Yes

5. Is the manuscript presented in an intelligible
fashion and written in standard English?

Reviewer #1: Yes

Reviewer #2: Yes

6. Review Comments to the Author

Reviewer #1: The authors have addressed my initial concern but I still have a few
comments and suggestions.

The authors added the statement “In this study, we initially confirmed our previous
findings of rapid increase in the number of CD8+ cells and the reversal of CD4/CD8
ratio in the peripheral blood of humanized mice following EBV infection.” to the
beginning of the results section. Please provide a reference.

The authors state the information in figure 1 is not an average bu a representation
from 1 mouse. I think showing the average with error bars would give a
representation of how consistent the results were.

Figure S1 is labeled in this order A,C, B, D. Is this intentional or is A, B, C, D
meant.

Figures 3 ,4 and 6 would be easier to read if labels such as A1, A2, B1, etc were
replaced with the antibody that was used for staining.

Figure 3 A1, A2. Are these the same picture at different magnifications? When
multiple pictures are shown for the the same antibody stain please indicate if it is
from the same mouse. The figures would be enhanced with a more descriptive labeling
scheme.

Reviewer #2: The introduction of additional images to Figures 2; 3; 4 and 6, were a
good improvement of the manuscript, strengthening the authors findings.

However, figure labeling could be improved to ensure better readability. The current
panel labeling (“A-1; B-1”) and the amount of panels in each figure, are hard for
readers to follow without constantly looking at the figure legend.

The authors could identify the panels with the molecule they are showing on top in
the vertical orientation while identifying the panels with the correct experimental
group in the horizontal. That would also potentially improve the figure legend
itself.

7. PLOS authors have the option to publish the peer
review history of their article (what does this mean?). If published, this will
include your full peer review and any attached files.

If you choose “no”, your identity will remain anonymous but your review may still be
made public.

**Do you want your identity to be public for this peer review?** For
information about this choice, including consent withdrawal, please see our
Privacy Policy.

Reviewer #1: No

Reviewer #2: **Yes: **Thiago Alves da Costa

---

## [Author Response · Author response to Decision Letter 1]

2 Mar 2021

Reviewer #1: The authors have addressed my initial concern but I still have a few
comments and suggestions.

The authors added the statement “In this study, we initially confirmed our previous
findings of rapid increase in the number of CD8+ cells and the reversal of CD4/CD8
ratio in the peripheral blood of humanized mice following EBV infection.” to the
beginning of the results section. Please provide a reference.

Response: Thank you for your helpful suggestions. We added a reference number at the
beginning of the results section properly.

The authors state the information in figure 1 is not an average bu a representation
from 1 mouse. I think showing the average with error bars would give a
representation of how consistent the results were.

Response: Based on your feedback, we revised figure 1 and showed the average values
of EBV-infected mice with error bars in the figure. 

Figure S1 is labeled in this order A,C, B, D. Is this intentional or is A, B, C, D
meant.

Response: As you pointed out, these labels ware inappropriate. We changed these
labels appropriately. A, C, B, D are incorrect, A, B, C, D are correct.

Figures 3 ,4 and 6 would be easier to read if labels such as A1, A2, B1, etc were
replaced with the antibody that was used for staining.

Response: As you pointed out, these labels were difficult to understand. A1, A2, B1,
etc in these labels ware replaced with the name of the stained antigen.

Figure 3 A1, A2. Are these the same picture at different magnifications? When
multiple pictures are shown for the the same antibody stain please indicate if it is
from the same mouse. The figures would be enhanced with a more descriptive labeling
scheme.

Response: Yes, images of A1 and A2 in the figure 3 are same picture at different
magnifications. Based on your feedback, we revised the figure and indicated mouse
numbers in the labels. 

Reviewer #2: The introduction of additional images to Figures 2; 3; 4 and 6, were a
good improvement of the manuscript, strengthening the authors findings.

However, figure labeling could be improved to ensure better readability. The current
panel labeling (“A-1; B-1”) and the amount of panels in each figure, are hard for
readers to follow without constantly looking at the figure legend.

The authors could identify the panels with the molecule they are showing on top in
the vertical orientation while identifying the panels with the correct experimental
group in the horizontal. That would also potentially improve the figure legend
itself.

Response: Thank you for your favorable comments and helpful suggestions. As you
pointed out, these figures were difficult to understand. Based on your feedback, we
showed the names of the stained antigens and mouse number at the top and side of
their panels in these figures.

to Reviewers.docx
---

## [Editor Report · Decision Letter 2]

17 Mar 2021

Human osteoclastogenesis in Epstein-Barr virus-induced erosive arthritis in humanized
NOD/Shi-scid/IL-2Rγnull mice

PONE-D-20-19201R2

Dear Dr. Takei,

We’re pleased to inform you that your manuscript has been judged scientifically
suitable for publication and will be formally accepted for publication once it meets
all outstanding technical requirements.

Kind regards,

Luwen Zhang

Academic Editor

PLOS ONE

---

## [Editor Report · Acceptance letter]

25 Mar 2021

PONE-D-20-19201R2 

Human osteoclastogenesis in Epstein-Barr virus-induced erosive arthritis in humanized
NOD/Shi-*scid/IL-2Rγ*^null^ mice
**

Dear Dr. Takei:

I'm pleased to inform you that your manuscript has been deemed suitable for
publication in PLOS ONE. Congratulations! Your manuscript is now with our
production department. 

Kind regards, 

on behalf of

Dr Luwen Zhang 

Academic Editor

PLOS ONE